# Dual heterogeneous structures lead to ultrahigh strength and uniform ductility in a Co-Cr-Ni medium-entropy alloy

X. H. Du[1,2], W. P. Li[1], H. T. Chang[2], T. Yang[1], G. S. Duan[2], B. L. Wu[2], J. C. Huang [1,3 ✉], F. R. Chen[1], C. T. Liu[1], W. S. Chuang[3], Y. Lu[4], M. L. Sui [4] & E. W. Huang[5]

Alloys with ultra-high strength and sufficient ductility are highly desired for modern engineering applications but difficult to develop. Here we report that, by a careful controlling alloy composition, thermomechanical process, and microstructural feature, a Co-Cr-Ni-based medium-entropy alloy (MEA) with a dual heterogeneous structure of both matrix and precipitates can be designed to provide an ultra-high tensile strength of 2.2 GPa and uniform elongation of 13% at ambient temperature, properties that are much improved over their counterparts without the heterogeneous structure. Electron microscopy characterizations reveal that the dual heterogeneous structures are composed of a heterogeneous matrix with both coarse grains (10~30 μm) and ultra-fine grains (0.5~2 μm), together with heterogeneous L1$_2$-structured nanoprecipitates ranging from several to hundreds of nanometers. The heterogeneous L1$_2$ nanoprecipitates are fully coherent with the matrix, minimizing the elastic misfit strain of interfaces, relieving the stress concentration during deformation, and playing an active role in enhanced ductility.

[1] Department of Materials Science and Engineering, Hong Kong Institute for Advanced Study, City University of Hong Kong, Kowloon, Hong Kong. [2] School of Materials Science and Engineering, Shenyang Aerospace University, Shenyang, China. [3] Department of Materials and Optoelectronic Science, National Sun Yat-Sen University, Kaohsiung, Taiwan. [4] Institute of Microstructure and Property of Advanced Materials, Beijing University of Technology, Beijing, China. [5] Department of Materials Science and Engineering, National Chiao Tung University, Hsinchu, Taiwan. ✉email: chihuang@cityu.edu.hk

Pursuing ultra-high strength (UHS > 2.0 GPa) metallic materials with sufficient uniform tensile strain (>8%) has long been a key for most challenged structural applications, such as aircraft landing gear, rocket cases, high-performance shafts and tubes, high-strength fasteners, and others[1,2]. The goal has been occasionally accomplished in maraging steels, in which strengthening mechanisms are through martensitic transformation and precipitations strengthening[3–5]. As to other ultra-strong structural materials used in more severe environment, such as Co-rich superalloys (MP35N or MP159), are also designed on the metallurgical basis of martensitic transformation occurring on cooling pure Co to a temperature below ~420 °C[6]. In such a case, effective strengthening species such as stacking faults (SFs), twins, as well as ε martensite can be easily introduced via planar slip of dislocations during the thermo-mechanical processes. However, as a matter of fact, the lamellar ε martensite are usually formed owing to the low stability of face-centered-cubic (FCC) phase of Co-rich superalloys[7–9]. The lamellar ε martensite usually degrades remarkably plastic deformation ability, because they strongly arrest the mobile dislocations causing the happening of pre-mature fracture[9,10].

As the above statements, to surmount the severe trade-off of strength and ductility, one feasible way is to control the stability of FCC phase in Co-rich alloys to avoid the untimely appearance of lamellar ε martensite during thermo-mechanical processes. Fortunately, a new concept of alloy system, referred as high-entropy alloys (HEAs) or medium-entropy alloys (MEAs), in which multiple principal elements are adopted to form single-phase structure with high symmetry can be employed to design the Co-rich alloy with stable FCC phase[11,12]. As a new class of materials, the properties of HEAs/MEAs are derived not from a dominant constituent but rather from multiple principal elements, and thus presenting great potential for unique combination of mechanical response compared with conventional alloys[13,14]. Here, ternary Co-Cr-Ni MEAs are promising candidates owing to their stable FCC phase and outstanding mechanical properties[15–17]. Furthermore, it is encouraged that some recent researches on the Co-Cr-Ni MEAs showed that dramatic enhancement in tensile yielding strength and remarkable tensile ductility can be achieved by architecting gradient hierarchical grains[18,19]. Nevertheless, in consideration of heterogeneous grains can only provide limited strengthening effects[18,19], enlightened by the ultra-strong maraging steels[4], other effective reinforcements are necessary to be introduced in achieving UHS. Recently, there are intensified studies in strengthening FCC-structured multi-principal element alloys (MPEAs) via precipitation strengthening and several researches[20–24] have demonstrated nano-scaled γ′ particles with L1$_2$ structure are especially effective reinforcement in achieving ultra-high mechanical properties.

In this study, to obtain a stable FCC matrix with low SF energy, we designed a Co-Cr-Ni-based MEA with an increasing Co content and decreasing Ni content compared with the equiatomic Ni-Co-Cr ternary alloy; 3 al.% Al and Ti were added to form fully coherent L1$_2$ precipitates. The heterogeneous grain structure of the FCC matrix was induced by the cryo-rolling (CR; 77 K) process and high-temperature annealing (900 °C/1 h), whereas the high-temperature annealing and subsequent aging (700 °C/4 h) introduce a heterogeneous precipitation. The dual heterogeneous structures of the alloy play a predominant role on the strengthening effect. The respectable ductility and good work-hardening capacity are attributed to the high dislocation density and fully coherent interface between the FCC matrix and the L1$_2$ precipitates. The strategy utilized in the current work, comprising composition design and thermomechanical process design, opens a new avenue for the development of promising heavy-duty structural materials.

## Results

**Tensile properties.** Figure 1a shows the tensile engineering stress–strain curves of our designed MEAs (Co: 34.46, Cr: 32.12, Ni: 27.42 with Al: 3, Ti: 3 (in at%)) subjected to three different treatments, namely: (1) CR only, (2) a CR followed by a high-temperature annealing (CRA), and (3) a CR followed by hybrid treatments of CRA and an subsequent aging (CRAA). The details of materials design and process procedures are presented in Methods and Supplementary Note 1. The tensile curve of the CR alloy shows a high yield strength $\sigma_y$ ~ 1.6 GPa and ultimate tensile strength $\sigma_u$ ~ 1.7 GPa with an inferior tensile uniform elongation $\varepsilon_{ue}$ ~ 2%. The high $\sigma_y$ indicates that the designed alloy exhibits a high-working hardening effect during CR. However, it is seen that the unstable flow of stress developed immediately after the yielding, suggesting it is at the plastic strain limit. Nonetheless, the CRA alloy shows that the alloy demonstrates mechanical properties with high $\sigma_y$ ~ 1.5 GPa, $\sigma_u$ ~ 1.7 GPa, and $\varepsilon_{ue}$ ~ 20%. This is encouraging, as the $\varepsilon_{ue}$ has been improved by ten times with a slight expense in $\sigma_y$ by the aid of an annealing after CR. The heterogeneous microstructure would induce heterogeneous deformation, resulting in variation of the mechanical strength, from 1.6 to 2.5 GPa. One of the best-performing samples achieves a $\sigma_y$ ~ 2.0 GPa and $\sigma_u$ ~ 2.2 GPa with a $\varepsilon_{ue}$ up to ~13%. This CRAA alloy shows a very high hardening response with an increase in $\sigma_y$ by about 0.5 GPa and practically no sharp reduction in ductility upon aging strengthening.

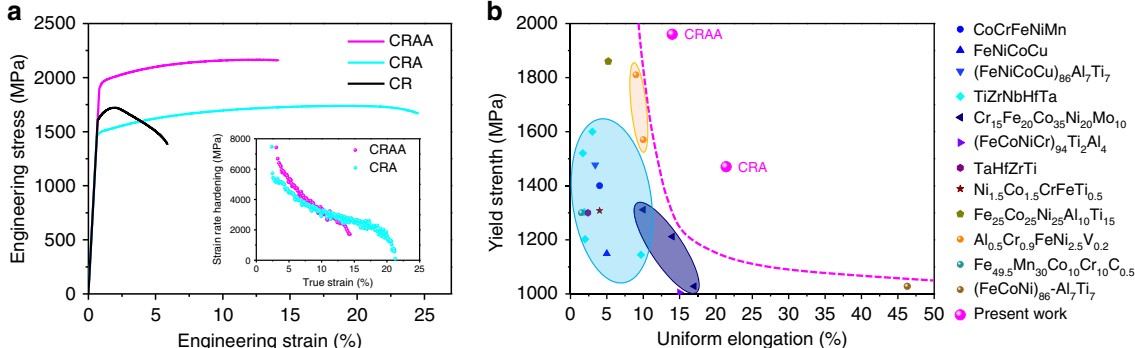

**Fig. 1 Mechanical properties of our alloys with different treatment states at 25 °C. a** Tensile curve of CR, CRA, and CRAA alloys, respectively. The insert in **a** presents the work-hardening rate ($d\sigma/d\varepsilon$) of CRA- and CRAA-treated alloys. **b** Maps of $\varepsilon_{ue}$ vs. $\sigma_y$ of high strength MEAs and HEAs with FCC or BCC matrix. The data of the mechanical properties of these reported materials are acquired from Supplementary Table 1.

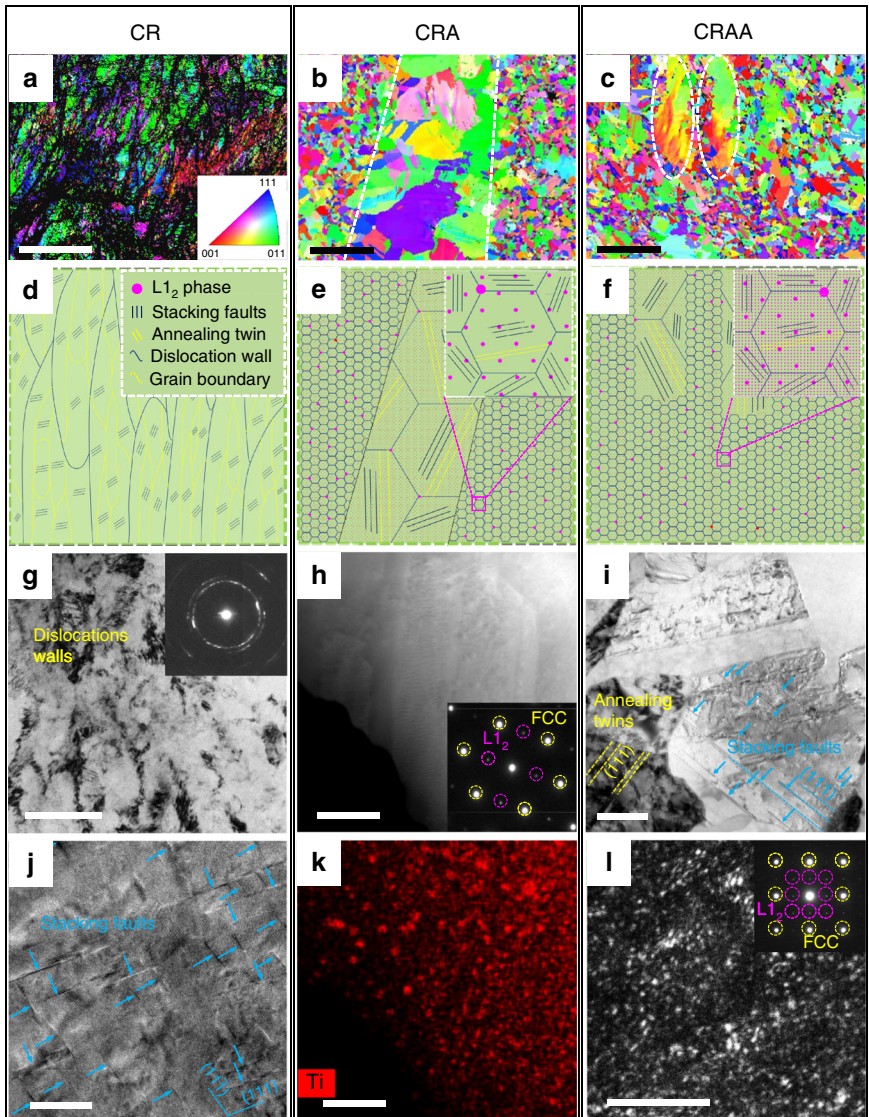

**Fig. 2 The microstructures of alloys subject to the CR, CRA and CRAA processes. a, d, g, j** For CR; **b, e, h, k** for CRA; and **c, f, i, l** for CRAA. **a–c** Normal direction EBSD inverse pole figure maps of alloys subject to CR, CRA, and CRAA, respectively. **b, c** Reveal non-uniform recrystallized microstructure resulting from the strain gradient during CR. The regions outlined by white dashed lines are coarse recrystallized grains (10∼30 μm), whereas other regions are fine recrystallized grains (0.5∼2 μm). **d, e, f** Schematic diagrams illustrating the microstructure evolution. Inserts of **e** and **f** are the enlarged images of fine recrystallized grains outlined by magenta dashed squares in **e** and **f**, respectively. **g** Transmission electron microscopy (TEM) bright-field (BF) image and SAED pattern revealing high-density dislocation walls and large numbers of fine dislocation cells in the CR alloy. **j** High-resolution TEM (HRTEM) image of deformed grains interior showing nano-spaced SFs (blue arrows) and widely existed dislocation locks in the CR alloy. **h, k** High-angle annular dark-field scanning transmission electron microscopy (HAADF-STEM) image and corresponding electron dispersive spectrometry (EDS) map about the Ti distribution exhibiting the Ti-rich particles; the inserted SAED patterns in **h** confirming that these particles in the CRA alloy are the L1$_2$ phase. **i** TEM-BF image of partially recrystallized microstructure showing numbers of annealing twins (yellow dashed lines) and SFs (blue arrows) exist in the CRAA alloy. **l** TEM dark-field (DF) image and SAED pattern revealing that there are L1$_2$-type particles with a high number density and an average diameter lower than 5 nm precipitating in the CRAA alloy. The scale bars in **a–c, g–l** are 10 μm, 20 μm, 20 μm, 200 nm, 500 nm, 200 nm, 20 nm, 500 nm, and 20 nm, respectively.

The combination of $\sigma_y$, $\sigma_u$, and $\varepsilon_{ue}$ indicates the current CRAA-treated MEA is apparently superior to those already reported FCC or body-centered cubic (BCC) structured multiple-principal-element alloys, as shown in Fig. 1b and Supplementary Fig. 1. For example, the CRAA-treated alloy gives promising characteristics, about 10% higher in $\sigma_y$, 15% higher in $\sigma_u$, and 44% higher in $\varepsilon_{ue}$ as compared with the ultra-strong HEA strengthened by high-content coherent nanoprecipitates (the orange region shown in Fig. 1b and Supplementary Fig. 1)[22]. The work hardening rate plot for the samples of annealing and aging can also be seen in the insert of Fig. 1a. Both the CRAA and

CRA conditions show high strain-hardening rates at high stress levels.

**Microstructures.** The electron microscopy images showing the microstructures via the CR, CRA, or CRAA treatment are presented in the left, middle, and right columns of Fig. 2, respectively. Figure 2a–c are the electron backscatter diffraction (EBSD) maps showing the evolution of grain structure for three alloys treated by CR, CRA, and CRAA, respectively. Figure 2d–f are the corresponding schematic version to show the structural evolution.

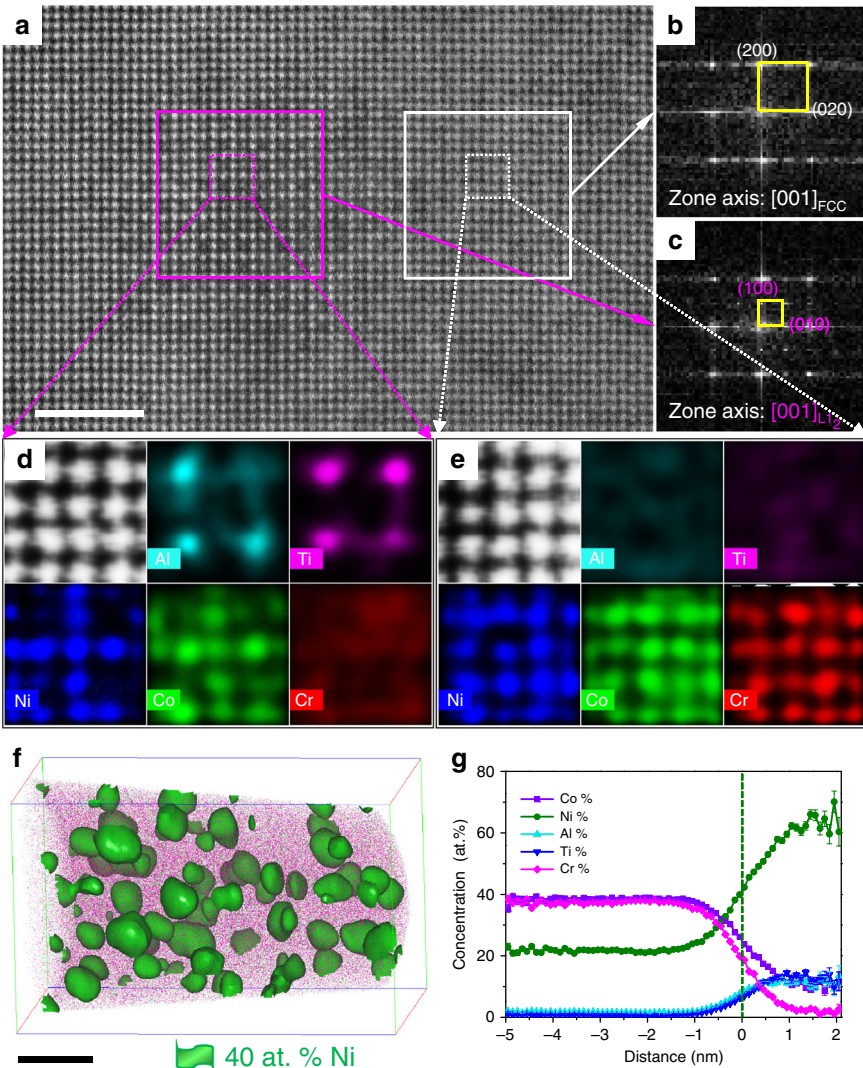

**Fig. 3 The microstructure of the CRAA alloy. a** The atomic-resolution HAADF-STEM image taken from [001] zone axis. Scale bar, 2 nm. **b**, **c** FFT patterns corresponding to areas outlined by magenta and white solid squares in **a**, respectively. **d**, **e** Atomic-resolution EDS maps corresponding to areas outlined by magenta and white dashed squares in **a**, respectively. **f** The three-dimensional reconstruction of 40 at% Ni iso-concentration surfaces presenting the morphologies of the fine particles (with an average diameter of 4.14 ± 0.62 nm) and matrix, respectively. Scale bar, 10 nm. **g** One-dimensional concentration profile showing the element distributions from matrix to Ni-enriched particle. Error bars, SD.

Figures 2g–l are the magnified views of one local area of Fig. 2a–c. After multi-pass CR, the CR alloy shows a typical microstructure of lamellae elongated grains containing dislocation cells (Fig. 2a, gg and high-density $(6.40 \times 10^{15} \, \text{m}^{-2})$ SFs (Fig. 2j) along the metal flow direction. Both of the CRA and CRAA alloys show heterogeneous structures for both grains and precipitates (Fig. 2b, c and Supplementary Fig. 2). For the CRA alloy, its heterogeneous grain structure containing coarse grains $(10 \sim 30 \, \mu\text{m})$ and ultra-fine $(0.5 \sim 2 \, \mu\text{m})$ grains. It is seen that both kinds of FCC-structured grains contain large numbers of nanotwins (e.g., Fig. 2b, c). In addition, the heterogeneous precipitates in the CRA alloy contains some nano-sized $(\sim 100 \, \text{nm})$ L1$_2$ phase spherical precipitates at grain boundaries and finer $(20 \sim 50 \, \text{nm})$ homogeneously dispersed within the FCC matrix (Fig. 2h, k and Supplementary Figs. 3 and 4a).

For the CRAA alloy, the EBSD map indicates that the grain sizes of coarse grains and ultra-fine grains do not increase apparently (Fig. 2c) as compared with those of the CRA alloy. However, TEM observation confirms that a high number of annealing twins $(2.97 \times 10^{12} \, \text{m}^{-2})$ and SFs (density = $1.83 \times 10^{14}$

$\text{m}^{-2}$, namely, mean spacing = 74 nm) survived in the matrix (Fig. 2i). The aging process introduces an extremely high number density of fine class of L1$_2$ ordering phases (<5 nm) in the FCC matrix (Fig. 2l and Supplementary Fig. 4b). The atomic resolution high-angle annular dark-field scanning TEM (HAADF-STEM) image taken from the [001] axis reveals that these L1$_2$ particles are perfectly coherent with matrix (Fig. 3a–c). Further atomic-resolution electron dispersive spectrometry (EDS) maps taken from precipitates reveal that Ti and Al atoms occupy the vertices of the L1$_2$ phases with a close-packed A$_3$B-type crystal structure, whereas the faced centers of the close-packed A$_3$B-type crystal structure are occupied by Ni and Co atoms (Fig. 3d). In contrast, the matrix has a random distribution of all elements (Fig. 3e). This is in good agreement with the atom probe tomography (APT) observation (Fig. 3f). The average size and the number density of these L1$_2$ phases were calculated from the APT result to be 4.14 ± 0.62 nm and $3.4 \times 10^{24} \, \text{m}^{-3}$, respectively. All information revealed by EDS maps is consistent with the contrast of Fig. 3a. In addition, APT characterization results also provide us with the precise composition of (Ni$_{87}$Co$_{13}$)$_3$(Al$_{50}$Ti$_{50}$) for the L1$_2$

precipitate phases (Fig. 3g). The results from the fitting procedure of synchrotron X-ray diffraction (SXRD) data (Supplementary Fig. 5) suggest that the lattice parameters of the recrystallization-annealed FCC matrix and the L1$_2$-type (Ni,Co)$_3$(Al,Ti) precipitated phase are 3.5629 ± 0.0003 Å and 3.5633 ± 0.0003 Å, respectively, with very low lattice mismatch about 0.011%.

## Discussion

Based on the above microstructural observations, the formation of dual heterogeneous structures and the associated underlying mechanisms responsible for the superior mechanical response are discussed below.

In this study, it is of interest to observe that the heterogeneous FCC-structured matrix comprised of coarse grains (10 ~ 30 μm) and ultra-fine grains (0.5 ~ 2 μm) formed during annealing at 900 °C/1 h in CRA process due to deformation gradients[15,18]. The grain structure is preserved after a subsequent aging at 700 °C/4 h in CRAA process mainly is due to the pinning effect caused by the intergranular L1$_2$ particles and the solute drag effect contributed by solutes of Al and Ti[25,26]. It is noted that the heterogeneous matrix due to partial recrystallization can create extra hetero-deformation induced (HDI) strengthening effect[18,27,28], owing to the accumulation of geometrically necessary dislocations in soft coarse grains. In addition, TEM observations shows that a high-density SFs with an average spacing of 74 nm were inherited in the CRAA alloy (Fig. 2i). The preservation of SFs in our case should be partially benefited by the low SF energy of the CoCrNi based matrix[16,29]. The high-density SFs formed during deformation can decrease the free path of dislocations and can cause the dynamic Hall–Petch effect at the same time to contribute to the working hardening response[24,30,31].

The second heterogeneous structure is the formation of the L1$_2$ precipitates with a wide range in size from several nanometers to hundreds of nanometers during 900 °C/1 h annealing and subsequent 700 °C/4 h aging processes. Previous reports showed that the L1$_2$-type precipitates in conventional superalloys or HEAs are mostly present in much lower number density (~10$^{22}$ m$^{-3}$) and coarser size (~10–50 nm)[20–24,32]. However, in this study, a heterogeneous distribution of L1$_2$-type precipitates with total 24.2% in volume fraction has been achieved by combined annealing and subsequent aging treatments in the designed alloy. We observe that besides grain-boundary precipitation, precipitating behavior within the grains is also extensive during 900 °C/1 h annealing. The total L1$_2$ volume fraction after 900 °C/1 h annealing can reach up to 13.25%. Furthermore, subsequent 700 °C/4 h aging introduces homogeneously precipitated extra L1$_2$ particles with a high number density (about 3.24 × 10$^{24}$ m$^3$) and small size (about 3 ~ 5 nm) within the FCC-structured matrix, as shown in Fig. 3. The volume fraction of these tiny L1$_2$ phase is 10.95% in the aged alloys (making the total L1$_2$ volume fraction of 24.2%). This structure provides a strong second-hardening effect, as presented in Supplementary Note 2.

Low lattice misfit (0.011%) and low heat-treatment temperature (700 °C/4 h) were two critical elements introducing homogeneously precipitated L1$_2$ particles with a high-density and small size during the aging process. On one hand, the low elastic-misfit energy brings a low-energy barrier for the homogeneous nucleation of L1$_2$ phase[4]. Thus, the nuclei of the L1$_2$ phase can precipitate in the matrix with a high number density. On the other hand, the low lattice misfit would decrease the specific interface free energy of the FCC/L1$_2$ interface, thus reducing the driven force for competitive coarsening and favoring the particles to maintain the near-spherical shape[33]. In addition, the slow diffusion rate of elements, especially Co element, at this low heat-treatment temperature can also avoid the premature onset of

Ostwald ripening[34,35]. As a result, extremely fine particles with an extremely high number density have been precipitated in an explosive mode in the current FCC matrix.

The current alloy exhibits exciting mechanical properties with a high $\sigma_y$ of ~2.0 GPa and $\sigma_u$ of ~2.2 GPa at room temperature after the final aging at 700 °C for 4 h. From the above microstructural observations, for the currently ultra-strong CoCrNiAlTi alloy, potential strengthening mechanisms should involve combination of solid-solution hardening $\Delta\sigma_s$, HDI hardening $\Delta\sigma_{HDI}$, and precipitation hardening $\Delta\sigma_p$, in addition to its lattice friction strength $\sigma_i$. For simplicity, we here apply the simplest linear addition rule, namely

$$\sigma_y = \sigma_i + \Delta\sigma_s + \Delta\sigma_{HDI} + \Delta\sigma_p. \qquad (1)$$

It has been confirmed that $\sigma_i$ of equiatomic CoCrNi is 218 MPa[36]. The strengthening contributions from solid-solution hardening, HDI hardening, and precipitation hardening are calculated to be ~49 MPa[21], ~800 MPa[18], and ~1115 MPa, respectively, (as presented in Supplementary Note 2), the total yield strength estimated by Eq. (1) would be 2182 MPa, in reasonable agreement with the measured 2000 MPa.

Another feature for the designed alloy is that a promising uniform elongation of ~13% is still achieved even at the strongly strengthened state. It is believed that this behavior is also originated from pronounced work-hardening rates induced during the tension process. Specifically, the strain-hardening rate curve shows that the specimens perform high strain-hardening rates at high stress levels (Fig. 1a). To correlate the work-hardening behavior with the underlying deformation mechanisms upon uniaxial tensile loading, the microstructural evolution of aging alloy after deformation is observed by SXRD and TEM.

From Supplementary Fig. 5 and Fig. 4, both the SXRD data and TEM observations confirm that no twins or hexagonal close-packed (HCP) phase is formed after deformation even under the ultra-high tensile stress level (~2.0 GPa). Furthermore, TEM observation shows only space-intersected SF and high-density dislocations can be observed in the fractured sample (Fig. 4a, b). This might come from the extremely high number density of L1$_2$ particles, causing the critical stress for the formation of nanotwins and HCP to increase sharply, as evidenced by other authors[21,23,30]. In the current case, the average spacing of the fractured sample between SFs is 49 nm (4.34 × 10$^{14}$ m$^{-2}$ for the density of SFs) (Fig. 4a), which is lower than that of the CRAA alloy before tensile testing. More importantly, the Lomer–Cottrell locks, formed by the reaction of two partials from two dissociated dislocations, are sessile and can not only strengthen the alloys by acting as barriers but also propagate more dislocations by serving as Frank–Read dislocation sources, providing a high dislocation

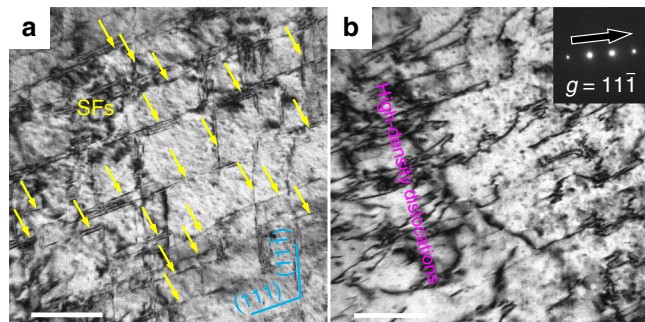

**Fig. 4 Microstructure of the CRAA sample after tensile testing.** a TEM BF image taken from near ⟨110⟩ axis, showing nano-spaced SFs (yellow arrows). **b** TEM BF image showing high density dislocations. Scale bars in **a** and **b** are both 100 nm.

density ($3.83 \times 10^{14} m^{-2}$) to accommodate subsequent plastic deformation[37], as shown in Fig. 4b. The tangle of high-density dislocations can bring considerable strain hardening effect and deformation capability before failure (Supplementary Fig. 6).

Furthermore, it has to mention the function of high-density nanoprecipitates distributed homogeneously in the FCC matrix on the improved deformation ability of this alloy. The size of ultra-fine $L1_2$ phase and the lattice misfit between the ultra-fine $L1_2$ phase and the FCC matrix are both one order of magnitude lower than the counterpart of $(FeCoNi)_{86}$-$Al_7Ti_7$, reported very recently as an epoch-making alloy[23]. The ultra-low elastic interfacial strain from the fully coherent interface can avoid the accumulation of dislocations near the interface. Therefore, the stress concentration around the precipitated particles can be relieved and the uniform plastic deformation can be maintained[4]. Also, it has been confirmed that the partial replacement of Ni atoms with Co atoms could significantly increase the ductility of $L1_2$ phases by lowering the covalent band directionality and the replacement of Al atoms with Ti atoms could decrease the environmental embrittlement by reducing the Al content[38].

Overall, by deliberately designing the compositions and processing method, an CoCrNi-AlTi alloys has been fabricated successfully and the combined properties are superior to existing FCC or BCC structured HEAs/MEAs. The CR, annealing, and aging processes make the alloy to develop dual heterogeneous nanostructures composing of heterogeneous partially recrystallized structure and heterogeneous $L1_2$ precipitates. This designed microstructure is regarded to be responsible for the significant enhancement in strength and the good tensile ductility at room temperature. As a result, an combination of UTS of 2.2 GPa and tensile uniform elongation of 13% has been achieved. The alloy elements and the thermomechanical treatments can be readily accepted by industry as promising heavy-duty structural materials.

## Methods

**Alloy fabrication**. Ingots of MEAs with the predetermined compositions ($Co_{34.46}Cr_{32.12}Ni_{27.42}Al_3Ti_3$, in at%) were prepared in a vacuum induction furnace using pure metals (purity greater than 99.9 wt%). The as-cast ingots with dimensions of $10 \times 10 \times 50\ mm^3$ were homogenized at 1200 °C for 2 h in an Ar atmosphere and subsequently hot-rolled at 1100 °C with a rolling reduction ratio of 50% (thickness changed from 10 mm to 5 mm). After the hot-rolling, alloys were homogenized again at 1200 °C for 2 h in an Ar atmosphere followed by water-quenching. After that, a multi-pass CR of the as-rolled alloys was carried out up to a total reduction ratio of 80% (minor strain with 0.05 mm per rolling). Before and immediately after each pass, the samples were immersed in a liquid nitrogen bath for 5 min. The cryo-rolled samples were annealed at 900 °C for 1 h followed by water quenching. Subsequently, the annealed samples were isothermally aged at 700 °C for 4 h followed by water quenching.

**Mechanical properties test**. Dog-bone-shaped specimens with a gauge reduced parallel length of 12.5 mm and a cross-section area of $3.2 \times 1.0\ mm^2$ were fabricated along the longitudinal direction of CR, CRA, and CRAA strip by electro-discharge machining for tensile testing. According to ASTM E8M standard, Uniaxial tensile tests were carried out at ambient temperature using a universal testing machine (GOTECH Al-7000-LA20, Taiwan) at a constant strain rate of $1 \times 10^{-3}\ s^{-1}$. At least three samples were tested to ensure the data reproducibility.

**Microstructure characterization**. EBSD measurements were carried out in a field-emission SEM (JEOL–JSM–7001 F) equipped with an automatic orientation acquisition system (Oxford Instruments-*HKL* Channel 5). The EBSD specimens were mechanically ground and polished, and then electro-polished with an electrolyte composed of 90% ethanol and 10% perchloric acid at room temperature. TEM characterizations were conducted on a JEOL JEM-2100 F instrument. STEM images and EDS maps were acquired on Thermo Fisher Titan $G^2$60-300 S/TEM (fitted with a high-brightness field-emission gun (X-FEG), probe Cs corrector and super X EDS with four windowless silicon drift detectors). STEM-HAADF images were taken using an annular-type detector with a collection angle ranging from 76 mrad to 200 mrad. EDS maps were collected and processed by the auto filter in the Esprit software. TEM and STEM samples with dimensions of $\Phi3\ mm \times 0.5\ mm$ were sliced by Struers cutting machine and then thinned to $50 \sim 60\ \mu m$ using

variant grit silicon carbides. After mechanical thinning, these samples were subject to a precise dimple grind for further thinning and polishing. Finally, ion milling was carried out on a cold stage (about −50 °C) at 5 keV, 5° until perforation and then at 2.5 keV, 3° for 10 min to reduce the thermal damage and surface amorphization.

APT was performed with a Cameca local electrode atom probe (CAMEACA LEAP 5000XR) under a voltage-pulsed mode. A specimen temperature of 50 K, a pulse repetition rate of 200 kHz, and a pulse fraction of 0.2 were used for APT measurements. Imago Visualization and Analysis Software version 3.6 was used for three-dimensional reconstructions and compositional analyses. Synchrotron X-ray radiation was also applied to examine structural evolutions during the aging treatment and deformation process, which were performed on the 11-ID-C beam line of the Advanced Photon Source, Taiwan. A monochromatic X-ray beam with an energy of 115 keV (with wavelength 0.010801 nm) was used. We carefully checked the lattice parameter of the solid solution treated and the aged samples by measuring the position of each peak in the synchrotron scattering X-ray data. The lattice mismatch between the matrix and precipitates was then estimated by the equation $\delta = 2(\alpha_{L12} - \alpha_{matrix})/(\alpha_{matrix} + \alpha_{L12})$, where $\alpha$ refers to the respective lattice parameter of each phase.

## Data availability
The data that support the findings of this study are available from the corresponding author upon reasonable request.

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

## Acknowledgements

We acknowledge the sponsorship from City University of Hong Kong under the projects 9380088, 7005078, and 9380092. The APT research was conducted in CityU, supported by CityU grant 9360161 and GRF grant C1027-14E.

## Author contributions

The initial alloy design were conducted by X.H.D., J.C.H., T.Y., and C.T.L. The thermochemical treatments were conducted by X.H.D., H.T.C., G.S.D., and B.L.W. C.T.L. and X.H.D. was in charge of the APT work. W.P.L., F.R.C., M.L.S., T.Y., W.S.C., and Y.L. took charge of the TEM details. E.W.H. was in charge of the XRD experiment. X.H.D., H.T.C., G.S.D., and B.L.W. conducted the mechanical testing. The strengthening analyses were done by X.H.D. and J.C.H. The manuscript writing was mainly conducted by X.H.D., J.C.H., and W.P.L.

## Competing interests

The authors declare no competing interests.
