## [Peer Review File · Nature Communications]

Reviewers' Comments:

Reviewer #1:

Remarks to the Author:

The manuscript presents a well designed alloy with a clever and complex processing leading to a remarkable combination of strength and tensile ductility. Due to extrem alloying costs and difficulties in industrial scale cryo-cold rolling of fcc materials due to high strain hardening I think there will be no market for such an alloy. So the main intention of the manuscript might be to present a novel approach or to provide guidelines to improve properties of other materials. The combination of alloying and different processing steps might be novel, but the individual steps are well known:

Especially the high strain hardening rate is remarkable and one of the main reasons for the high uniform elongation at such high stress levels:

In principal, in fcc materials it is possible to store a high density of dislocations and especially the reduction of the stacking fault energy (here by high Co content) in a NiCr alloy will decrease the amount of dynamic recovery and will lead to high strain hardening rate. The stacking faults will add to strain hardening. But this is well understood.

A broad grain size distribution will contribute to a high strain hardening rate and there are some publications of this in the field of ultrafine grained steels. In the manuscript it is described as HDI strengthening effect (Page 7 line 151-153: "It is noted that the hierarchical matrix due to partial recrystallization can create extra hetero-deformation induced (HDI) strengthening effect owing to the induced back and forward stresses.") This is an important point, but not novel (also already published in quotation 7 for a high-entropy alloy). The explanation in quotation 7 for this effect is also better than in the current mauscript by using the GND density concept.

A material with a dispersion of very fine coherent particles have a high tendency for localized deformation like shear bands when the coherent particles are sheared by dislocations (local softening). This would reduce ductility. It might be that the larger L12-precipitates here leads to Orowan looping and inducing local strain hardening which might compensate the local softening of the small particles. The autors assume that the large and small precipitates are all sheared by dislocations: Supplementary information (SI) page 2 line 63-64: "Due to the near fully-coherent relationship of γ/γ' and extremely high number density, the reinforced γ' phase is substantially sheared by the gliding dislocations during the deformation." I have my doubts about this: At least the large γ' precipitates might be circumvented by dislocations (Orowan mechanism) when the stress to cut throught is higher then the stress for for Orowan looping. This is maybe not very important in calculating the flow stress, but might be important for avoiding localization of plastic flow, increasing strain hardening and should be discussed to explain the "magic" of the "hierachical" microstructure. There is a good publication of this problem (Modeling of the yield strength of a stainless maraging steel, R. Schnitzer, S. Zinner and H. Leitner, Scripta Materialia 62 (2010) 286–289) (I am not involved in this publication).

I think that the level of novelty is not high enough to justify a publication in Nature com. But for scientist working in the field of high- and medium-entropy alloys the properties and approaches in the manuscript are interesting. So the manuscript should be submitted to a more specialized magazine.

I add more points were a correction or modification of the manuscript is required:

Discussion why L12 precipitates are small and high in number density:

"Low lattice misfit (0.011%) and low heat treatment temperature (700oC/4 h) were two critical elements introducing homogeneously precipitated L12 particles with a high-density and small size during the aging process. On one hand, the low elastic misfit energy brings a low energy barrier for the homogeneous nucleation of L12 phase¹⁶. So, the nuclei of the L12 phase can precipitate in the matrix with a high number density."

I fully agree.

" On the other hand, the low elastic misfit energy decreases the interface formation energy of the FCC/L12 interface, thus reducing the driven force for competitive coarsening and inducing the particles to maintain the near-spherical shape¹⁷.

I do not agree. Typically in nucleation theory the formation of new interfaces and for coherent interfaces, an additional term, the elastic distortion has to be taken into account. Here the authors (and the authors of quotation 17) sum this up as interface energy. This is dangerous: During growth the elastic distortion might rise with precipitate size and even coherency can be lost. Therefore a low elastic misfit might favor Ostwald ripening (not much elastic penalty against precipitate growth).

"Additionally, the low heat treatment temperature decreases the diffusion rate of elements, which can also avoid the premature onset of Ostwald ripening."

I agree. But it should be mentioned that the higher chemical driving force at lower temperature (higher supersaturation) favors a high nucleation number density.

For easier alloying cost calculations and comparison with similar alloys would be good to give the alloy composition not only in atomic percent, but also in mass percent (maybe in form of a table.).

Page 2 line 62-63 "Both alloys show multiple strain hardening stages at high-stress levels." Better to state that ""Both conditions, CRAA and CRA, show multiple strain hardening stages at high-stress levels." because CRAA and CRA are the same alloy (same composition). But I see only one strain hardening stage and no multiple strain hardening stages at high-stress levels. The minimum in strain hardening rate is just the transition from elastic to elastic/plastic and might indicate a very weak tendency for localized deformation (Lüders band).

Would be an improvement to draw an additional true stress versus true strain curve for CRAA and CRA in the diagram with strain hardening rate versus true strain in the inset Fig. 2a. This would allow to judge if the material fails by plastic instability (necking) or by damage.

Page 3 line 81: "...high-density ($6.40 \times 10^{15} \text{m}^{-2}$)..." unit is wrong, do you mean " $6.40 \times 10^{15} \text{m}^{-2}$)?" Fig. 2: typo in Fig.2d "Stacking faults"

Caption of Fig. 2 is misleading ("The microstructure evolution of alloys subject to ..."), better "The microstructure of the alloy subjected to..." because evolution might confuse what happened later during tensile testing and there is only one alloy.

Page 5 line 108-109 "...there are high-density L12-type particles ..." this is misleading (specific weight of particles?) better "...there are L12-type particles with high number density ..."

Page 5 line 116: "The aging process introduces extremely high number density ..." better ""The aging process introduces an extremely high number density ..."

Page 7 line 153-154: "Page 7 line 151-153 "In addition, TEM observations shows that high-density stacking faults with ..." better "In addition, TEM observations shows that a high density of stacking faults with ..."

Page 7 line 155: "The reservation of stacking faults ..." I think you mean "The preservation of stacking faults ..."

Page 7 line 156-158: "These residual stacking faults can impede dislocations movement by decreasing the mean free path of dislocations and increase work-hardening response by promoting the so-called dynamic Hall-Petch effect^{10,11}."

Page 8 line 204: "This might come from the extremely high-density L12 particles,..." improve English, for example : "This might come from the extremely high number density of L12 particles,..."

Page 8 line 207: " ($4.34 \times 10^{14} \text{m}^{-2}$ for the density of SFs), unit is wrong

Page 8 line 207: " ... Frank-Reed ..." correct to : " ... Frank-Read ..."

Page 8 line 212: " ... providing high-density dislocations ($3.83 \times 10^{14} \text{m}^{-2}$) ..." change to " ... providing a high dislocation density ($3.83 \times 10^{14} \text{m}^{-2}$) ..." (also correct unit as shown). Similar

corrections in line 213 (high-density dislocations).

Supplementary information (SI)

Page 1 line 8-9: "To suppress the recrystallized process owing to the increased apparent activation energy of recrystallization process." Improve English, for example: "To suppress recrystallization owing to the increased apparent activation energy for recrystallization."

Page 1 line 8-9: "To promote the precipitating of γ' phase via decreasing ..." Improve English, for example: "To promote the precipitation of γ' phase via decreasing ..."

Page 1 line 10-12: "In this study, the moderate contents of 3 at% Al and 3 at% Ti are added to precipitate the γ' phase during the aging process in order to avoid the formation of brittle BCC phase." A little unclear, can be improved to "In order to avoid the formation of brittle BCC phase, only moderate contents of 3 at% Al and 3 at% Ti are added to precipitate the γ' phase during the aging process."

page 2 line 64: "... the reinforced γ' phase" is misleading, I think the authors mean "... the reinforcing γ' phase" or maybe better "... the precipitation strengthening γ' phase".

Page 5 figure caption: "Fig. S3. APT results of the annealing alloy." better "Fig. S3. APT results of the annealed alloy."

Reviewer #2:

Remarks to the Author:

The authors have investigated a near-equiatomic Co-Cr-Ni base alloy to which 3 at.% each of Al and Ti were added to form L12 structured (ordered) Ni₃Al-type intermetallic precipitates in a f.c.c. matrix. The alloy was vacuum induction melted and cast, following which it was "homogenized" at 1200 C for 2 h, hot rolled at 1100 C (50% thickness reduction), and "homogenized" again at 1200 C for 2 h. Microstructures of the material after these heat treatments were not provided, so it is impossible to judge whether the relatively short 2-h anneals were sufficient to result in true homogenization. Some of the heterogeneities that they observe after later processing might well be due to insufficient homogenization. The casting was then cryo-rolled (after immersing in liquid nitrogen) for an 80% thickness reduction. This state is referred to as the CR state. Some of the CR material was annealed at 900 C for 1 h to produce the CRA material, and the CRA material was then annealed at 700 C for 4 h to produce the CRAA material.

As expected, the CR material was relatively strong and brittle because of the heavy cold work. After the first anneal, there was precipitation of relatively coarse (20-100 nm) L12 precipitates in the CRA material, which would be expected to increase strength, but this was offset by recovery and recrystallization which resulted in a decrease in strength and thus an increase in ductility. The second anneal resulted in additional formation of much finer (5 nm) L12 precipitates in the CRAA material resulting in a significant increase in strength but at the expense of ductility. Broadly speaking, these are the expected trends when taking into account the effects of precipitates, cold work, recovery, and recrystallization, on strength and ductility. The authors have not identified any unique (or previously unrecognized) mechanisms of deformation in their alloy that make this paper scientifically impactful. Previous papers by some of the current authors have ploughed similar territory in alloys with similar chemistries. Therefore, it is recommended that the paper be submitted to a more specialized journal after the following points have been addressed.

The CRA material had a central band of relatively coarse grains surrounded by much finer grains (as shown in the EBSD image). It was not clear from the description in the paper whether this central band was in the center of the rolled sheet, whether there was only one such band in the cross-section, or whether there were more bands. It is somewhat surprising that after 80% cold work there remained strong deformation gradients (as the authors suggest). Without knowing the cause of this heterogenous grain structure (temperature gradient during cryo rolling? composition gradient due to insufficient homogenization?), it will be impossible to replicate it in another alloy

system.

It is not clear why order hardening is not dependent on precipitate size (eq. 3 in the SI). This equation is from Ardell, *Met. Trans.* 16A, 2131-2165, 1985. Are the assumptions that Ardell used to arrive at this equation (eq. 85b in his paper) satisfied in the authors' alloy? If one looks at the derivation of this equation, precipitate size is present in some of the earlier equations (84a and 84b) but drops out when certain assumptions are made. If one is in the precipitate cutting regime, shouldn't larger precipitates be more difficult to shear than smaller ones?

An important missing piece in this work is a careful comparison with a control alloy in which the grain size is uniform and the precipitate size is uniform. What would happen, for example, if after cryo rolling, the material was directly annealed at 700 C for 4 h? Presumably, this would result in one population of very fine precipitates, which can be used to test the idea that the two different populations are indeed necessary to obtain the observed mechanical properties.

The paper is also missing other relevant information. For example, from the images in Fig. 2b-c, it is not clear whether CRA and CRAA are fully recrystallized or only partially recrystallized. If the latter is the case, then strengthening from the residual dislocation content would have to be considered. There is also a high density of stacking faults in CRA and CRAA. What is their contribution to strength? The same question for the annealing twins. There are also significant differences in texture for the three cases shown in Fig. 2a-c. What is the effect of texture strengthening?

In Fig. 2a, the elastic strain is around 2.5% (artifact of machine compliance). This needs to be subtracted from the engineering strain to obtain the uniform elongation. The values reported in the manuscript (at least for the CRAA case) are artificially high as a result.

In Fig. 3f, why was the 40 at% Ni iso-concentration surface used to delineate the precipitates? The Ni concentration increases sharply (over a couple of nm) from around 20% in the matrix to ~65% in the precipitate. It would appear to make more sense to use a 60 or 65% Ni iso-surface. How does the choice of the iso-concentration surface affect parameters such as number density, volume fraction, and size of precipitates?

Line 85: Both large and small grains supposedly contain large numbers of nanotwins (referring to Figs. 2b,c). However, they were not obvious to me. Perhaps the authors can mark them somehow? Are they deformation twins from the prior cryo rolling or annealing twins? If the former, such a supposedly high density of nanotwins should have a marked effect on strength, but it was not accounted for in the strength analysis. If the latter, I have difficulty understanding why they would be nanoscale in dimension because typically there should not be more than one or two twins per recrystallized grain.

The stacking fault densities are reported to 2 decimal places. What is the precision with which this can be measured?

Line 193: How is the solid solution strengthening contribution calculated separately from the intrinsic lattice friction? Extrapolation of the Hall-Petch plot to infinite grain size cannot deconvolute the solid solution effect from the intrinsic lattice friction (for any given alloy).

The original papers should be cited for SI equations 1-3 (Ardell) rather than some of the current team's recent work. The same holds for the APB energy (in previous work by some of the current authors, the original literature was indeed cited, which is the correct practice).

Reviewer #3:

Remarks to the Author:

This manuscript entitled "Superb strength-ductility synergy in medium-entropy alloy by engineering dual hierarchical structures" authored by Du et al. reports the material design of an ultrastrong yet ductile medium-entropy $\text{Co}_{34.46}\text{Cr}_{32.12}\text{Ni}_{27.42}\text{-Al}_3\text{Ti}_3$ (at %) alloy that possesses a dual hierarchical structure composed of a hierarchical partially-recrystallized grain structure along with hierarchical L12 precipitations ranging from several to hundreds of nanometers. The dual hierarchical structure leads to the achievement of a superior combination of tensile strength of 2.2 GPa and uniform elongation of 14% in room temperature, which is unprecedented in the existing fcc/bcc high/medium entropy alloys. Specifically, the multiscale hierarchical structure supplies several strengthening mechanisms simultaneously (i) hetero-deformation induced strengthening (HDI) effect ascribed to the profound geometric necessary dislocations induced by the incompatible deformation between larger and smaller grains, and (ii) dynamic Hall-Petch effect promoted by the abundant residual stacking faults originated from the lowered stacking faulty energy in the deviated alloy composition ($\text{Co}_{34.46}\text{Cr}_{32.12}\text{Ni}_{27.42}$, in at %) from the equiatomic state. The hierarchical precipitates are fully coherent with the matrix, reducing the elastic misfit strain and contributing to the enhanced ductility. Overall, the strength and ductility in the precipitation-hardened heterogeneous CoCrNi-AlTi alloy is very impressive. However, despite these impressive results, the manuscript itself is somewhat lacking and in many respects not befitting the results. Specifically, the following points might need the authors' attention.

1. Page 2, first paragraph: it is unusual that the article starts directly with the experimental results without a first paragraph to introduce the background and motivation.
2. An elongated lamellar structure is formed in all the three alloys processed after cryo-rolling (CR), cryo-rolling plus 900 °C annealing for 1 h (CRA), and cryo-rolling plus 900 °C annealing for 1 h followed by 700 °C aging for 4 h (CRAA). However, it is not indicated whether the tensile properties were measured in the loading direction along the longitudinal direction of the lamellar grains or not. In addition, the method to measure the engineering strain should be specified.
3. The strain hardening curves of the CRA and CRAA alloys presented in the inset of Fig. 1a show three featured stages: (1) macro-yielding, where the strain hardening rate drops significantly; (ii) stage II, where strain hardening rate rises slightly; and (iii) a third stage where the strain hardening rate decreases continuously. Can the authors provide any explanation for the occurrence of the second stage? Is there any specific deformation mechanism associated with the rise of the strain hardening rate after the macro-yielding?
4. The existence of two hierarchical structures at both the grain scale and precipitation scale in the CRAA alloy clearly plays a critical role in strengthening the alloy. Qualitative imaging and schematic illustrations of the dual hierarchical structures are presented in Fig. 2. Is it possible to statistically quantify the size distribution of the grains and the precipitates?
5. Page 8, lines 207-208, "the average spacing of the fractured sample between SFs is 48.6 nm ($4.34 \times 10^{14} \text{ m}^2$ for the density of SFs) (Fig. 4a), which is evidently lower than that of the CRAA alloy before tensile testing": can the authors elaborate the reason why the average spacing between SFs was increased after tension? Is the reduction of SF-spacing in deformed microstructure related to the strengthening mechanism or the ductilization mechanism?
6. The caption to Fig. 4 is missing.

Reply Letter

Manuscript Ref. No. NCOMMS-19-539430-T

Title: Superb strength-ductility synergy in medium-entropy alloy by engineering dual hierarchical structures

Journal: Nature Comm

11 February, 2020

Dear editor and reviewers:

Thank you very much for the review. The reviewers' constructive comments can greatly strengthen our paper. We have tried our best in replying or adding new information in text. The point to point replies are described below.

Reviewers' comments:

Reviewer #1 (Remarks to the Author):

1. The manuscript presents a well designed alloy with a clever and complex processing leading to a remarkable combination of strength and tensile ductility. Due to extreme alloying costs and difficulties in industrial scale cryo-cold rolling of fcc materials due to high strain hardening, I think there will be no market for such an alloy. So the main intention of the manuscript might be to present a novel approach or to provide guidelines to improve properties of other materials.

The combination of alloying and different processing steps might be novel, but the individual steps are well known: Especially the high strain hardening rate is remarkable and one of the main reasons for the high uniform elongation at such high stress levels. In principal, in fcc materials it is possible to store a high density of dislocations and especially the reduction of the stacking fault energy (here by high Co content) in a NiCr alloy will decrease the amount of dynamic recovery and will lead to high strain hardening rate. The stacking faults will add to strain hardening. But this is well understood.

Reply: Thanks a lot for your positive comments.

2. A broad grain size distribution will contribute to a high strain hardening rate and there are some publications of this in the field of ultrafine grained steels. In the manuscript it is described as HDI strengthening effect (Page 7 line 151-153: "It is noted that the hierarchical matrix due to partial recrystallization can create extra hetero-deformation induced (HDI) strengthening effect owing to the induced back and forward stresses.") This is an important point, but not novel (also already published in quotation 7 for a high-entropy alloy). The explanation in quotation 7 for this effect is also better than in the current manuscript by using the GND density concept.

Reply: Thanks a lot. We agree that the HDI strengthening effect in high/medium-entropy alloy is an important point, though might not be so novel. The combination of different kinds of strengthening mechanisms in the medium-entropy alloy is a new trial. Moreover, the extremely fine $L1_2$ precipitates (less than 5 nm) formed during the 700 °C/4 h aging process is another innovation. Additionally, we agree with the opinion in quotation 7 that the accumulation of geometrically necessary dislocations (GNDs) in soft coarse grains is the key reason for the HDI strengthening effect. On one hand, the accumulated GNDs can directly

strengthen the material through forest hardening (Cross slip is difficult to be activated in this alloy due to the lower stacking fault energy). On the other hand, the long-range stresses induced by the density gradient of GNDs can impede the dislocation motion in regions away from interfaces and cause additional hardening. The existence of back and forward stresses is just a stress performance of the alloy. So we change our text and attribute the HDI strengthening effect to the accumulation of GNDs in soft coarse grains, as described in p. 8.

3. A material with a dispersion of very fine coherent particles have a high tendency for localized deformation like shear bands when the coherent particles are sheared by dislocations (local softening). This would reduce ductility. It might be that the larger L1₂-precipitates here leads to Orowan looping and inducing local strain hardening which might compensate the local softening of the small particles. The authors assume that the large and small precipitates are all sheared by dislocations: Supplementary information (SI) page 2 line 63-64: “Due to the near fully-coherent relationship of γ/γ' and extremely high number density, the reinforced γ' phase is substantially sheared by the gliding dislocations during the deformation.” I have my doubts about this: At least the large γ' precipitates might be circumvented by dislocations (Orowan mechanism) when the stress to cut through is higher than the stress for Orowan looping. This is maybe not very important in calculating the flow stress, but might be important for avoiding localization of plastic flow, increasing strain hardening and should be discussed to explain the “magic” of the “hierachical” microstructure. There is a good publication of this problem (Modeling of the yield strength of a stainless maraging steel, R. Schnitzer, S. Zinner and H. Leitner, Scripta Materialia 62 (2010) 286–289) (I am not involved in this publication).

Reply: Thanks a lot. We have now read the paper you recommended. According to the approach proposed by this paper to describe the age-hardening phenomena, we calculated the critical radius of the dislocation-precipitates interaction transformed from shear to bypass. However, we found that if we put the volume fraction of the L1₂ precipitate in CRAA alloy ($f = 24.2\%$) into the Equation 3 in the Scripta paper, we encounter an abnormal result that the distance between particles ($L = (1.23(2\pi/(3f))^{1/2} - 2(2/3)^{1/2})r$) are equal to 1.985r (smaller than 2r). In other words, there is no channel for dislocation slip, which is in conflict with the experimental results. So the Equation 3 in the Scripta paper is appropriate for the calculation of precipitation hardening induced by Orowan mechanism when the volume fraction of precipitates is lower than 24%.

Fortunately, we found another paper [He J Y, Wang H, Wu Y, et al. Precipitation behavior and its effects on tensile properties of FeCoNiCr high-entropy alloys. Intermetallics, 79 (2016) 41-52], talking about the interaction mechanism of the L1₂ precipitate. In this paper, another equation was used to calculate the critical stress $\Delta\sigma_{\text{orowan}}$ via the Orowan mechanism:

$$\Delta\sigma_{\text{orowan}} = M \cdot 0.4Gb / (\pi \sqrt{1-\nu} \cdot \ln(2\bar{r}/b)) / L_p \quad (1)$$

where the Taylor factor $M=3.06$ is a constant for polycrystalline FCC structure, $G = 77$ GPa is the shear modulus of the matrix (taken from conventional Ni-based alloys), $\nu = 0.31$ is the Poisson ratio for the (FeCoNiCr)₉₄Ti₂Al₄ (at. %) alloy, and $b = \frac{\sqrt{2}}{2}a = 0.252$ nm is the Burgers vector of the dislocations in CRAA alloy.

In this paper, we have confirmed that if we have assumed that dislocations can shear all L1₂ particles, the strengthening effect is derived from the ordering strengthening. Namely,

$$\Delta\sigma_{\text{shear}} = \Delta\sigma_{\text{ordering}} = 0.81M \frac{\gamma_{\text{APB}}}{2b} \left(\frac{3\pi f}{8} \right)^{1/2} \quad (2)$$

where $\gamma_{APB} = 0.3 \text{ J/m}^2$ is the antiphase boundaries energy.

In this work, the critical radius for the L₁₂ precipitate in CRAA alloy was calculated by iterative changing the particle radius until the stresses obtained from Equations (1) and (2) delivered the same value. The calculated result of the critical radius is 65.1 nm, which is higher than the radius of most of L₁₂ particles precipitated in grains. So we consider that the shearing mechanism is the predominant mechanism of the precipitation hardening effect, agreeing well with our experimental observations.

4. Discussion why L₁₂ precipitates are small and high in number density:

Reply: Thanks a lot. The cause of small size and high number density of the precipitates is a result of multiple reasons. Firstly, very small lattice misfit of the coherent matrix/particle interface, resulting in a very low driving force for particle nucleation, resulting in high number density [Jiang, S. et al. Ultrastrong steel via minimal lattice misfit and high-density nanoprecipitation. *Nature* 544 (2017) 460-464]. Secondly, the lattice diffusion for Co-rich high entropy alloy is also relatively lower than conventional alloys [Zhao, Y. et al. Exceptional nanostructure stability and its origins in the CoCrNi-based precipitation-strengthened medium-entropy alloy. *Materials Research Letters* 7 (2019) 152-158]. Thirdly, the cryo-rolling process with large thickness reduction for high entropy alloy is unique in this work, favoring the storage of defects accumulated during cryo-rolling and the stimulation of high-density precipitation of the L₁₂ phase.

5. “Low lattice misfit (0.011%) and low heat treatment temperature (700°C/4 h) were two critical elements introducing homogeneously precipitated L12 particles with a high-density and small size during the aging process. On one hand, the low elastic misfit energy brings a low energy barrier for the homogeneous nucleation of L12 phase. So, the nuclei of the L12 phase can precipitate in the matrix with a high number density.”

I fully agree.

“ On the other hand, the low elastic misfit energy decreases the interface formation energy of the FCC/L12 interface, thus reducing the driven force for competitive coarsening and inducing the particles to maintain the near-spherical shape¹⁷.

I do not agree. Typically, in nucleation theory the formation of new interfaces and for coherent interfaces, an additional term, the elastic distortion has to be taken into account. Here the authors (and the authors of quotation 17) sum this up as interface energy. This is dangerous: During growth, the elastic distortion might rise with precipitate size and even coherency can be lost. Therefore, a low elastic misfit might favor Ostwald ripening (not much elastic penalty against precipitate growth).

“Additionally, the low heat treatment temperature decreases the diffusion rate of elements, which can also avoid the premature onset of Ostwald ripening.”

I agree. But it should be mentioned that the higher chemical driving force at lower temperature (higher supersaturation) favors a high nucleation number density.

Reply: Thanks a lot. For the *Ostwald ripening* process after precipitation from the supersaturated solid solution during the annealing process, the coarsening rate is given by the expression [C. Wagner, *Z. Elektrochem.*, 65 (1961) 581]:

$$r_t^3 - r_0^3 = 8D\sigma V_m C_\alpha(\infty)t/9RT, \quad (3)$$

where D is the volume diffusion coefficient of the diffusing species and σ is the specific surface free energy of the particle/matrix interface.

Here, the element diffusion rates of are delayed by the low heat treatment temperature plus the sluggish kinetic nature of high-entropy alloys. And the specific surface free energy of the

particle/matrix interface is low due to the low lattice misfit (0.011%). Hence, the coarsening rate is low.

We should change the original sentence “On the other hand, the low elastic misfit energy would decrease the interface formation energy of the FCC/L1₂ interface, thus reducing the driven force for competitive coarsening and inducing the particles to maintain the near-spherical shape” to “On the other hand, the low lattice misfit would decrease the specific surface free energy of the FCC/L1₂ interface, thus reducing the driven force for competitive coarsening and favoring the particles to maintain the near-spherical shape” in p. 8.

6. For easier alloying cost calculations and comparison with similar alloys would be good to give the alloy composition not only in atomic percent, but also in mass percent (maybe in form of a table.).

Reply: Thanks a lot. We have added a Table in the Supplement.

Elements	Co	Cr	Ni	Al	Ti
Atomic percent	34.46	32.12	27.42	3.00	3.00
Mass percent	36.69	30.07	29.08	1.46	2.60

7. Page 2 line 62-63 “Both alloys show multiple strain hardening stages at high-stress levels.” Better to state that “Both conditions, CRAA and CRA, show multiple strain hardening stages at high-stress levels.” because CRAA and CRA are the same alloy (same composition). But I see only one strain hardening stage and no multiple strain hardening stages at high-stress levels. The minimum in strain hardening rate is just the transition from elastic to elastic/plastic and might indicate a very weak tendency for localized deformation (Lüders band). Would be an improvement to draw an additional true stress versus true strain curve for CRAA and CRA in the diagram with strain hardening rate versus true strain in the inset Fig. 2a. This would allow to judge if the material fails by plastic instability (necking) or by damage.

Reply: Thanks a lot. The true strain-true stress curves for CRAA and CRA are presented below, showing that continuous strain hardening is maintained during the whole plastic deformation stage. It suggests that the high strain hardening rates are responsible for the remarkable plasticity (uniform elongations) for the current materials at such a high-strength level. Moreover, no multiple strain hardening stages at high-stress levels are observed, the materials always fail by damage.

Also, the reviewer has put forward a good suggestion to correct the error. The original sentence of “Both alloys show multiple strain hardening stages at high-stress levels” has been changed to “Both the CRAA and CRA conditions show high strain hardening rates at high-stress levels” in p. 3.

The true strain-true stress curve for CRAA and CRA.

8. Page 3 line 81: "...high-density ($6.40 \times 10^{15} \text{m}^{-2}$)..." unit is wrong, do you mean " $6.40 \times 10^{15} \text{m}^{-2}$)?"

Reply: Thanks a lot. Yes, the reviewer is right. We have corrected it in p. 4.

9. Fig. 2: typo in Fig.2d "Staking faults"

Caption of Fig. 2 is misleading ("The microstructure evolution of alloys subject to ..."), better "The microstructure of the alloy subjected to..." because evolution might confuse what happened later during tensile testing and there is only one alloy.

Reply: Thanks a lot. We have corrected it in p. 5.

10. Page 5 line 108-109 "...there are high-density L12-type particles ..." this is misleading (specific weight of particles?) better "...there are L12-type particles with high number density ..."

Reply: Thanks a lot. We have corrected it as suggested in p. 6.

11. Page 5 line 116: "The aging process introduces extremely high number density ..." better "The aging process introduces an extremely high number density ..."

Reply: Thanks a lot. We have corrected it as suggested in p. 6.

12. Page 7 line 153-154: "Page 7 line 151-153 "In addition, TEM observations shows that high-density stacking faults with ..." better "In addition, TEM observations shows that a high density of stacking faults with ..."

Reply: Thanks a lot. We have corrected it as suggested in p. 8.

13. Page 7 line 155: "The reservation of stacking faults ..." I think you mean "The preservation of stacking faults ..."

Reply: Thanks a lot. We have corrected it as suggested in p. 8.

14. Page 7 line 156-158: "These residual stacking faults can impede dislocations movement by decreasing the mean free path of dislocations and increase work-hardening response by promoting the so-called dynamic Hall-Petch effect."

Reply: Thanks a lot. The high-density stacking faults formed during the deformation can decrease the free path of dislocations and cause the dynamic Hall-Petch effect at the same time to contribute to the working hardening response. These are described in p. 8.

15. Page 8 line 204: "This might come from the extremely high-density L12 particles,..." improve English, for example : "This might come from the extremely high number density of L12 particles,..."

Reply: Thanks a lot. We have corrected it as suggested in p. 9.

16. Page 8 line 207: " $(4.34 \times 10^{14} \text{m}^{-2})$ for the density of SFs), unit is wrong

Reply: Thanks a lot. We have corrected it as suggested in p. 9

17. Page 8 line 207: "... Frank-Reed ..." correct to : "... Frank-Read ..."

Reply: Thanks a lot. We have corrected it as suggested in p. 9.

18. Page 8 line 212: "... providing high-density dislocations ($3.83 \times 10^{14} \text{m}^{-2}$) ..." change to "... providing a high dislocation density ($3.83 \times 10^{14} \text{m}^{-2}$) ..." (also correct unit as shown).

Similar corrections in line 213 (high-density dislocations).

Reply: Thanks a lot. We have corrected it as suggested in p. 9.

Supplementary information (SI)

19. Page 1 line 8-9: “To suppress the recrystallized process owing to the increased apparent activation energy of recrystallization process.” Improve English, for example: “To suppress recrystallization owing to the increased apparent activation energy for recrystallization.”

Reply: Thanks a lot. We have corrected it as suggested in SI p. 1.

20. Page 1 line 8-9: “To promote the precipitating of γ' phase via decreasing ...” Improve English, for example: “To promote the precipitation of γ' phase via decreasing ...”

Reply: Thanks a lot. We have corrected it as suggested in SI p. 1.

21. Page 1 line 10-12: “In this study, the moderate contents of 3 at% Al and 3 at% Ti are added to precipitate the γ' phase during the aging process in order to avoid the formation of brittle BCC phase.” A little unclear, can be improved to “In order to avoid the formation of brittle BCC phase, only moderate contents of 3 at% Al and 3 at% Ti are added to precipitate the γ' phase during the aging process.”

Reply: Thanks a lot. We have corrected it as suggested in SI p. 1.

22. page 2 line 64: “... the reinforced γ' phase” is misleading, I think the authors mean “... the reinforcing γ' phase” or maybe better “... the precipitation strengthening γ' phase”.

Reply: Thanks a lot. We have corrected it as suggested in SI p. 2.

23. Page 5 figure caption: “Fig. S3. APT results of the annealing alloy.” better “Fig. S3. APT results of the annealed alloy.”

Reply: Thanks a lot. We have corrected it as suggested in SI p. 5.

Reviewer #2 (Remarks to the Author):

1. The authors have investigated a near-equiatomic Co-Cr-Ni base alloy to which 3 at.% each of Al and Ti were added to form L12 structured (ordered) Ni₃Al-type intermetallic precipitates in a f.c.c. matrix. The alloy was vacuum induction melted and cast, following which it was “homogenized” at 1200 C for 2 h, hot rolled at 1100 C (50% thickness reduction), and “homogenized” again at 1200 C for 2 h. Microstructures of the material after these heat treatments were not provided, so it is impossible to judge whether the relatively short 2-h anneals were sufficient to result in true homogenization. Some of the heterogeneities that they observe after later processing might well be due to insufficient homogenization. The casting was then cryo-rolled (after immersing in liquid nitrogen) for an 80% thickness reduction. This state is referred to as the CR state. Some of the CR material was annealed at 900 C for 1 h to produce the CRA material, and the CRA material was then annealed at 700 C for 4 h to produce the CRAA material.

Reply: Thanks a lot. We have conducted electron backscattered diffraction (EBSD) characterizations on the hot-rolled sample which was subsequently treated by the 2 h annealing at 1200°C. As shown in the Figures below, the EBSD contrast and inverse pole figure (IPF) maps show a fully recrystallized microstructure with the presence of coarse equiaxed grains and there is no obvious fine-grained area in the sample.

The microstructure of the hot-rolled sample which was subsequently treated by the 2 h annealing at 1200°C: (a) band contrast image and (b) EBSD contrast with its inverse pole figure (IPF).

2. As expected, the CR material was relatively strong and brittle because of the heavy cold work. After the first anneal, there was precipitation of relatively coarse (20-100 nm) $L1_2$ precipitates in the CRA material, which would be expected to increase strength, but this was offset by recovery and recrystallization which resulted in a decrease in strength and thus an increase in ductility. The second anneal resulted in additional formation of much finer (5 nm) $L1_2$ precipitates in the CRAA material resulting in a significant increase in strength but at the expense of ductility. Broadly speaking, these are the expected trends when taking into account the effects of precipitates, cold work, recovery, and recrystallization, on strength and ductility. The authors have not identified any unique (or previously unrecognized) mechanisms of deformation in their alloy that make this paper scientifically impactful. Previous papers by some of the current authors have ploughed similar territory in alloys with similar chemistries.

Reply: Thanks a lot. The Co-rich compositions have increased the activation energy for recrystallization process, retarding the recrystallization during the high-temperature and low-temperature ageing processes to result in partially recrystallized heterogeneous grain structure. Furthermore, the cryo-rolling process with large thickness reduction is also unique in this work to stimulate the precipitation of $L1_2$ phase. In summary, by combining the cryo-rolling and two-step ageing process, the designed Co-rich HEA will promote the formation of the unique microstructure: the heterogeneously recrystallized grains and heterogeneous distribution of $L1_2$ -structure precipitates with a high density. Such an unusual dual heterogeneous structure enables the superior mechanical response of this newly designed alloy.

3. The CRA material had a central band of relatively coarse grains surrounded by much finer grains (as shown in the EBSD image). It was not clear from the description in the paper whether this central band was in the center of the rolled sheet, whether there was only one such band in the cross-section, or whether there were more bands. It is somewhat surprising that after 80% cold work there remained strong deformation gradients (as the authors suggest). Without knowing the cause of this heterogeneous grain structure (temperature gradient during cryo rolling? composition gradient due to insufficient homogenization?), it will be impossible to replicate it in another alloy system.

Reply: Thanks a lot. As EBSD characterization is difficult to ensure high resolution (for fine-grain regions) and large-area observation (to make it clear whether there was only one such band in the cross-section or whether there were more bands in CRA and CRAA alloys), we choose OM characterization to figure out this question. The microstructures of the CRA

and CRAA alloys are shown in the Figures below. It reveals that many bands are widely distributed in both CRA and CRAA alloys and surrounded by many fine grains.

Microstructural characterization for the (a) CRA and (b) CRAA alloys.

The heterogeneous grain structure in the CRA and CRAA alloys are easy to be replicated. During the cryo-rolling process, some grains with unique slip systems and orientations would experience greater plastic strain and possess higher dislocation density. These grains could grow remarkably faster than other grains during the post annealing process, thereby resulting in the formation of heterogeneous grain structure, namely, several coarse grains embedded in ultra-fine grains. There has been a publication which studied the influence of the orientation on the growth rate of grain [G. Wu, D.J. Jensen, Recrystallisation kinetics of aluminium AA1200 cold rolled to true strain of 2, Materials Science and Technology, 21 (2005) 12.]

4. It is not clear why order hardening is not dependent on precipitate size (eq. 3 in the SI). This equation is from Ardell, Met. Trans. 16A, 2131-2165, 1985. Are the assumptions that Ardell used to arrive at this equation (eq. 85b in his paper) satisfied in the authors' alloy? If one looks at the derivation of this equation, precipitate size is present in some of the earlier equations (84a and 84b) but drops out when certain assumptions are made. If one is in the precipitate cutting regime, shouldn't larger precipitates be more difficult to shear than smaller ones?

Reply: Thanks a lot. Yes, as replied for Question 3 of Reviewer 1, the fine ordered precipitates should be shearable by passing dislocations and the larger ones might be in the dislocation looping regime. We have re-analyzed the strengthening contribution by more careful calculations. The critical radius of the transformation from shearing mechanism to looping mechanism is 65.1 nm. Most of L_{12} particles precipitated in our present CRAA alloys are smaller than this critical radius.

5. An important missing piece in this work is a careful comparison with a control alloy in which the grain size is uniform and the precipitate size is uniform. What would happen, for example, if after cryo rolling, the material was directly annealed at 700 C for 4 h? Presumably, this would result in one population of very fine precipitates, which can be used to test the idea that the two different populations are indeed necessary to obtain the observed mechanical properties.

Reply: Thanks a lot. The mechanical response for the same alloy with homogeneous grain structures and precipitates, as well as the samples under different degrees of heterogeneous

grain structures and precipitate sizes are under research, collaborating with a senior visiting professor, who is the key scholar working on heterogeneous deformation and is currently visiting in City University of Hong Kong for half year. These new and complete results will be presented in another paper in future.

For the time being, we have already evaluated the mechanical properties of the alloy which was directly annealed at 700°C for 4 h after cryo-rolling process. The following figure shows the tensile engineering stress-strain curve of the designed alloy subjected to cryo-rolling and subsequent aging (700°C/4 h) process. The current alloy exhibits mechanical properties with a yield stress of 2.03 GPa and UTS about 2.25 GPa at room temperature. However, the tensile ductility is only ~5%. The true strain-strain hardening rate curve inserted in the figure shows that the alloy sustains relatively steady strain hardening behavior with a high hardening rate (over 3000 MPa) during the entire plastic straining stage. The combined mechanical properties are obviously inferior to that presented in our submitted manuscript.

Mechanical properties of the cryo-rolled $\text{Co}_{34.46}\text{Cr}_{32.12}\text{Ni}_{27.42}\text{Al}_3\text{Ti}_3$ samples with aging treatment state at ambient temperature

6. The paper is also missing other relevant information. For example, from the images in Fig. 2b-c, it is not clear whether CRA and CRAA are fully recrystallized or only partially recrystallized. If the latter is the case, then strengthening from the residual dislocation content would have to be considered. There is also a high density of stacking faults in CRA and CRAA. What is their contribution to strength? The same question for the annealing twins. There are also significant differences in texture for the three cases shown in Fig. 2a-c. What is the effect of texture strengthening?

Reply: Thanks a lot. Yes, the residual dislocations and stacking faults, and very minor twins, could contribute some to the strength. In this work, the annealing temperatures (900°C/1 h and 700°C/4 h) are high enough to decrease substantively the dislocations density by recrystallization process and restoration process (the formation of sub-grains). As a result, the remained stacking faults in CRA and CRAA contribute little to the whole strength because their wide spacing distance. The amounts of annealing twins are little, so their contribution to the whole strength is also tiny.

Recently, a concept, hetero-deformation induced (HDI) hardening was proposed in [Y. Zhu and X. Wu. Perspective on hetero-deformation induced (HDI) hardening and back stress. *Materials Research Letters* 7 (2019) 393-398]. They considered that the heterogeneous grain structure introduced by partial recrystallization could provide dramatic strengthening effect for alloys. Yang et al. [M. Yangm et al. Dynamically reinforced heterogeneous grain structure prolongs ductility in a medium-entropy alloy with gigapascal yield strength. *Proc.*

Natl. Acad. Sci. U.S.A., (2018) 201807817] has reported that the equal-atomic CoCrNi medium-entropy alloy reinforced by heterogeneous grain structure can achieve a high yield stress in excess to 1 GPa. Considering that there is neither second-phase precipitates nor other solute elements in this CoCrNi medium-entropy alloy, the strengthening effect (~800 MPa) is wholly attributed to the HDI effect. In this paper, as the grain size of CRA and CRAA alloy is similar with the equal-atomic CoCrNi medium-entropy alloy reported and also span from hundreds of nanometer to micrometer size, we cite their results of the HDI hardening effect (800 MPa) for an estimation.

To evaluate the texture strengthening, the pole figure and inverse pole figure of the CRAA are shown below, showing that no strong crystallographic-orientation is found to favor the formation of deformation twins in the CRAA sample. It is reasonable to conclude that there is no obvious strong texture strengthening in our material.

Pole figure and inverse pole figure for the CRAA.

7. In Fig. 2a, the elastic strain is around 2.5% (artifact of machine compliance). This needs to be subtracted from the engineering strain to obtain the uniform elongation. The values reported in the manuscript (at least for the CRAA case) are artificially high as a result.

Reply: Thanks a lot. Yes, the reviewer is right. The elastic modulus of CoCrNi equi-atomic medium-entropy alloy is about 226 GPa [S. Yoshida, et al. Friction stress and Hall-Petch relationship in CoCrNi equi-atomic medium entropy alloy processed by severe plastic deformation and subsequent annealing. Scripta Materialia 134 (2017) 33-36], while the elastic modulus of the L_{12} phase (Ni_3Al) is about 203 GPa [S.V. Prikhodko et al. Temperature and composition dependence of the elastic constants of Ni_3Al . Metallurgical and Materials Transactions A, 30 (1999) 2403-2408]. Considering the similar value between the CoCrNi matrix and L_{12} phase, we presume 220 GPa as the elastic modulus to correct the three tensile curves, as shown below. The tensile elongations of the CRA and CRAA alloys are about 23% and 14%, respectively. There are numerous tensile testing results, and the elongation is always around 20-25% for the CRA samples and about 13-18% for the CRAA samples.

Corrected tensile curve of CR, CRA and CRAA alloys, respectively.

8. In Fig. 3f, why was the 40 at% Ni iso-concentration surface used to delineate the precipitates? The Ni concentration increases sharply (over a couple of nm) from around 20% in the matrix to ~65% in the precipitate. It would appear to make more sense to use a 60 or 65% Ni iso-surface. How does the choice of the iso-concentration surface affect parameters such as number density, volume fraction, and size of precipitates?

Reply: Thanks a lot. The use of 40 at% Ni is judgement based on the change profiles of increasing Ni, Al, and Ti, as well as the decreasing Cr and Co. From these profiles, we set the middle point for all these five profiles as the origin (Distance = 0 nm) in Fig. 3g. If we use 60 or 65 at% for Ni to define the precipitate boundary, it might be in conflict with the middle point of the element profiles for other elements.

9. Line 85: Both large and small grains supposedly contain large numbers of **nanotwins** (referring to Figs. 2b,c). However, they were not obvious to me. Perhaps the authors can mark them somehow? Are they deformation twins from the prior cryo rolling or annealing twins? If the former, such a supposedly high density of nanotwins should have a marked effect on strength, but it was not accounted for in the strength analysis. If the latter, I have difficulty understanding why they would be nanoscale in dimension because typically there should not be more than one or two twins per recrystallized grain.

Reply: Thanks a lot. From Figs. 2b and 2c in the manuscript, it can be seen that the twins have appeared in the coarse grains. Obviously, they are annealing twins. The annealing nanotwins are difficult to be detected by EBSD due to the small thickness. Some annealing nanotwins are marked in Fig. 2i of the manuscript. Additionally, Zhao et al. [Y.L. Zhao et al. Heterogeneous precipitation behavior and stacking-fault-mediated deformation in a CoCrNi-based medium-entropy alloy. *Acta Materialia* 138 (2017) 72-82] have reported that the numerous annealing nanotwins were formed in the recrystallized grains, as shown below. In this work, the Co-rich alloys exhibit lower stacking faults energy, thus it is common for the formation of some multiple twins with nano-scale thickness during the annealing process.

TEM images of the single-phase CoCrNi MEA in the as-recrystallized state [Y.L. Zhao et al. Heterogeneous precipitation behavior and stacking-fault-mediated deformation in a CoCrNi-based medium-entropy alloy. *Acta Materialia* 138 (2017) 72-82]

10. The stacking fault densities are reported to 2 decimal places. What is the precision with which this can be measured?

Reply: Thanks a lot. It is difficult to tell the precision of the stacking fault densities. TEM characterization can only acquire the information from local regions. We just calculated the stacking fault densities from several TEM images taken from different locations.

11. Line 193: How is the solid solution strengthening contribution calculated separately from the intrinsic lattice friction? Extrapolation of the Hall-Petch plot to infinite grain size cannot deconvolute the solid solution effect from the intrinsic lattice friction (for any given alloy).

Reply: Thanks a lot. We used 49 MPa as the solid solution strengthening. This solid solution strengthening contribution (by Al and Ti) was derived from the experiment test which was reported in (CoCrNi)₉₄Al₃Ti₃ alloy [Y.L. Zhao, et al. Heterogeneous precipitation behavior and stacking-fault-mediated deformation in a CoCrNi-based medium-entropy alloy. *Acta Materialia* 138 (2017) 72-82]. The $\sigma_1=218$ MPa is the lattice friction strength of equiatomic CoCrNi alloys [S. Yoshida, et al. Friction stress and Hall-Petch relationship in CoCrNi equi-atomic medium entropy alloy processed by severe plastic deformation and subsequent annealing. *Scripta Materialia* 134 (2017) 33-36]. Considering there is a slight component deviation between equiatomic CoCrNi alloys and the matrix of our present alloys, we used and cited the lattice friction strength of equiatomic CoCrNi alloys.

12. The original papers should be cited for SI equations 1-3 (Ardell) rather than some the current team's recent work. The same holds for the APB energy (in previous work by some of the current authors, the original literature was indeed cited, which is the correct practice).

Reply: Thanks a lot. We have corrected it as suggested in SI p. 2.

Reviewer #3 (Remarks to the Author):

1. This manuscript entitled "Superb strength-ductility synergy in medium-entropy alloy by engineering dual hierarchical structures" authored by Du et al. reports the material design of an ultrastrong yet ductile medium-entropy Co_{34.46}Cr_{32.12}Ni_{27.42}-Al₃Ti₃ (at %) alloy that possesses a dual hierarchical structure composed of a hierarchical partially-recrystallized grain structure along with hierarchical L12 precipitations ranging from several to hundreds of nanometers. The dual hierarchical structure leads to the achievement of a superior

combination of tensile strength of 2.2 GPa and uniform elongation of 14% in room temperature, which is unprecedented in the existing fcc/bcc high/medium entropy alloys. Specifically, the multiscale hierarchical structure supplies several strengthening mechanisms simultaneously (i) hetero-deformation induced strengthening (HDI) effect ascribed to the profound geometric necessary dislocations induced by the incompatible deformation between larger and smaller grains, and (ii) dynamic Hall-Petch effect promoted by the abundant residual stacking faults originated from the lowered stacking faulty energy in the deviated alloy composition (Co_{34.46}Cr_{32.12}Ni_{27.42}, in at %) from the equiatomic state. The hierarchical precipitates are fully coherent with the matrix, reducing the elastic misfit strain and contributing to the enhanced ductility. Overall, the strength and ductility in the precipitation-hardened heterogeneous CoCrNi-AlTi alloy is very impressive. However, despite these impressive results, the manuscript itself is somewhat lacking and in many respects not befitting the results. Specifically, the following points might need the authors' attention.

Reply: Thanks for the positive comments.

2. Page 2, first paragraph: it is unusual that the article starts directly with the experimental results without a first paragraph to introduce the background and motivation.

Reply: Thanks a lot. We have now revised the Introduction Section, as shown below.

Pursuing ultrahigh strength (UTS>2.0 GPa) metallic materials with sufficient uniform tensile strain (UE>8%) has long been a key for most challenged structural applications, such as aircraft landing gear, rocket cases, high-performance shafts and tubes, high-strength fasteners, and others^{1,2}. The goal has been occasionally accomplished in maraging steels, in which strengthening mechanisms are through martensitic transformation and precipitations strengthening³⁻⁵. As to other ultra-strong structural materials used in more severe environment, such as Co-rich superalloys (MP35N or MP159), are also designed on the metallurgical basis of martensitic transformation occurring on cooling pure Co to a temperature below ~420°C. In such a case, effective strengthening species such as stacking faults, twins as well as ϵ martensite can be easily introduced via planar-slip of dislocations during the thermo-mechanical processes. However, as a matter of fact the lamellar ϵ martensite are usually formed owing to the low stability of facial-centered-cubic (FCC) phase of Co-rich superalloys⁶⁻⁸. The lamellar ϵ martensite usually degrades remarkably plastic deformation ability because they strongly arrest the mobile dislocations causing the happening of pre-mature fracture^{8,9}.

As above statements, to surmount the severe “trade-off” of strength and ductility, one feasible way is to control the stability of FCC phase in Co-rich alloys to avoid the untimely appearance of lamellar ϵ martensite during thermo-mechanical processes. Fortunately, a new concept of alloy system, referred as high entropy alloys (HEAs) or medium entropy alloys (MEAs), in which multiple principal elements are adopted to form single-phase structure with high symmetry can be employed to design the Co-rich alloy with stable FCC phase^{10,11}. As a new class of materials, the properties of HEAs/MEAs are derived not from a dominant constituent but rather from multiple principal elements and thus presenting great potential for unique combination of mechanical response compared with conventional alloys^{12,13}. Here, ternary Co-Cr-Ni MEAs are promising candidates owing to their stable FCC phase as well as outstanding mechanical properties¹⁴⁻¹⁶. Furthermore, it is encouraged that some recent researches on the Co-Cr-Ni MEAs showed that dramatic enhancement in tensile yielding strength and remarkable tensile ductility can be achieved by architecting gradient hierarchical grains^{17,18}.

Nevertheless, enlightened by the ultra-strong maraging steels, in consideration of hierarchical grains can only provide limited strengthening effects^{17,18}, other effective

reinforcements are necessary to be introduced in achieving ultra-high strength. Recently, there are intensified studies in strengthening FCC structured MPEAs via precipitation strengthening, and several researches¹⁹⁻²³ have demonstrated nano-scaled γ' particles with $L1_2$ structure are especially effective reinforcement in achieving ultra-high mechanical properties. Accordingly, in this study, Co-Cr-Ni-Al-Ti quintuple alloy is selected to realize the comprehensive strengthening by engineering the expected gradient hierarchical structures reinforced by nano-scaled $L1_2$ precipitates.

References

- 1 Inoue, J., Nambu, S., Ishimoto, Y. & Koseki, T. Fracture elongation of brittle/ductile multilayered steel composites with a strong interface. *Scripta Materialia* 59, 1055-1058 (2008).
- 2 Jr, M. & W., J. Maraging steels: Making steel strong and cheap. *Nature Materials* 16, 787-789 (2017).
- 3 Stiller, K., H?ttestrand, M. & Danoix, F. Precipitation in 9Ni-12Cr-2Cu maraging steels. *Acta Materialia* 46, 6063-6073 (1998).
- 4 Jiang, S. et al. Ultrastrong steel via minimal lattice misfit and high-density nanoprecipitation. *Nature* 544, 460-464, doi:10.1038/nature22032 (2017).
- 5 He, B. B. et al. High dislocation density-induced large ductility in deformed and partitioned steels. *Science* 357, 1029-1032, doi:10.1126/science.aan0177 (2017).
- 6 Asgari, S., El-Danaf, E., Shaji, E., Kalidindi, S. R. & Doherty, R. D. The secondary hardening phenomenon in strain-hardened MP35N alloy. *Acta Materialia* 46, 5795-5806, doi:10.1016/s1359-6454(98)00235-3 (1998).
- 7 Singh, R. P. & Doherty, R. D. Strengthening in MULTIPHASE (MP35N) alloy: Part I. ambient temperature deformation and recrystallization. *Metallurgical Transactions A* 23, 307-319 (1992).
- 8 Ishmaku, A. & Han, K. Characterization of cold-rolled and aged MP35N alloys. *Materials Characterization* 47, 139-148 (2001).
- 9 Shaji, E. M., Kalidindi, S. R., Doherty, R. D. & Sedmak, A. S. Fracture properties of multiphase alloy MP35N. *Materials Science & Engineering A* 349, 313-317 (2003).
- 10 Zhang, Y. et al. Microstructures and properties of high-entropy alloys. *Progress in Materials Science* 61, 1-93, doi:10.1016/j.pmatsci.2013.10.001 (2014).
- 11 Miracle, D. B. & Senkov, O. N. A critical review of high entropy alloys and related concepts. *Acta Materialia* 122, 448-511, doi:10.1016/j.actamat.2016.08.081 (2017).
- 12 Gludovatz, B. et al. A fracture-resistant high-entropy alloy for cryogenic applications. *Science* 345, 1153-1158, doi:10.1126/science.1254581 (2014).
- 13 Li, Z., Zhao, S., Ritchie, R. O. & Meyers, M. A. Mechanical properties of high-entropy alloys with emphasis on face-centered cubic alloys. *Progress in Materials Science* 102, 296-345, doi:10.1016/j.pmatsci.2018.12.003 (2019).
- 14 Gludovatz, B. et al. Exceptional damage-tolerance of a medium-entropy alloy CrCoNi at cryogenic temperatures. *Nat Commun* 7, 10602, doi:10.1038/ncomms10602 (2016).
- 15 Laplanche, G. et al. Reasons for the superior mechanical properties of medium-entropy CrCoNi compared to high-entropy CrMnFeCoNi. *Acta Materialia* 128, 292-303, doi:10.1016/j.actamat.2017.02.036 (2017).
- 16 Miao, J. et al. The evolution of the deformation substructure in a Ni-Co-Cr equiatomic solid solution alloy. *Acta Materialia* 132, 35-48, doi:10.1016/j.actamat.2017.04.033 (2017).
- 17 Yang, M. X. et al. Dynamically reinforced heterogeneous grain structure prolongs ductility in a medium-entropy alloy with gigapascal yield strength. *Proc. Natl. Acad. Sci. U. S. A.* 115, 7224-7229, doi:10.1073/pnas.1807817115 (2018).

- 18 Slone, C. E., Miao, J., George, E. P. & Mills, M. J. Achieving ultra-high strength and ductility in equiatomic CrCoNi with partially recrystallized microstructures. *Acta Materialia* **165**, 496-507, doi:10.1016/j.actamat.2018.12.015 (2019).
- 19 He, J. Y. *et al.* A precipitation-hardened high-entropy alloy with outstanding tensile properties. *Acta Materialia* **102**, 187-196, doi:10.1016/j.actamat.2015.08.076 (2016).
- 20 Zhao, Y. L. *et al.* Heterogeneous precipitation behavior and stacking-fault-mediated deformation in a CoCrNi-based medium-entropy alloy. *Acta Materialia* **138**, 72-82, doi:10.1016/j.actamat.2017.07.029 (2017).
- 21 Liang, Y. J. *et al.* High-content ductile coherent nanoprecipitates achieve ultrastrong high-entropy alloys. *Nat Commun* **9**, 4063, doi:10.1038/s41467-018-06600-8 (2018).
- 22 Yang, T. *et al.* Multicomponent intermetallic nanoparticles and superb mechanical behaviors of complex alloys. *Science* **362**, 933-937, doi:10.1126/science.aas8815 (2018).
- 23 Yang, T. *et al.* Nanoparticles-strengthened high-entropy alloys for cryogenic applications showing an exceptional strength-ductility synergy. *Scripta Materialia* **164**, 30-35, doi:10.1016/j.scriptamat.2019.01.034 (2019).

3. An elongated lamellar structure is formed in all the three alloys processed after cryo-rolling (CR), cryo-rolling plus 900°C annealing for 1 h (CRA), and cryo-rolling plus 900°C annealing for 1 h followed by 700°C aging for 4 h (CRAA). However, it is not indicated whether the tensile properties were measured in the loading direction along the longitudinal direction of the lamellar grains or not. In addition, the method to measure the engineering strain should be specified.

Reply: Thanks a lot. Yes, the tensile properties were all measured along the longitudinal direction of the lamellar grains, as indicated in SI p. 1. The engineering strain is measured based on the common equation of $\Delta L/L_0$, where $\Delta L = L_f - L_0$ and L_f is the final gauge length and L_0 is the initial gauge length.

4. The strain hardening curves of the CRA and CRAA alloys presented in the inset of Fig. 1a show three featured stages: (i) macro-yielding, where the strain hardening rate drops significantly; (ii) stage II, where strain hardening rate rises slightly; and (iii) a third stage where the strain hardening rate decreases continuously. Can the authors provide any explanation for the occurrence of the second stage? Is there any specific deformation mechanism associated with the rise of the strain hardening rate after the macro-yielding?

Reply: Thanks a lot. We rechecked the strain hardening curves again and found that the three featured stages shown in these curves seem to be caused by errors in data smoothing. The correct strain hardening curves of the CRA and CRAA alloys are re-presented below, showing that continuous strain hardening is maintained during the whole plastic deformation stage. It suggests that the high strain hardening rates are responsible for the remarkable plasticity (uniform elongations) for the current materials at such a high-strength level. Moreover, no multiple strain hardening stages at high-stress levels are observed, the materials always fail by damage.

The strain hardening curves of the CRA and CRAA alloys

The original sentence of “Both alloys show multiple strain hardening stages at high-stress levels” has been changed to “Both the CRAA and CRA conditions show high strain hardening rates at high-stress levels” in p. 3. The original image of the strain hardening curves for CRAA and CRA alloy in Fig. 1 has also been corrected.

5. The existence of two hierarchical structures at both the grain scale and precipitation scale in the CRAA alloy clearly plays a critical role in strengthening the alloy. Qualitative imaging and schematic illustrations of the dual hierarchical structures are presented in Fig. 2. Is it possible to statistically quantify the size distribution of the grains and the precipitates?

Reply: Thanks a lot. The mechanical response for the same alloy with homogeneous grain structures and precipitates, as well as the samples under different degrees of heterogeneous grain structures and precipitate sizes are under research, collaborating with a senior visiting professor, who is the key scholar working on heterogeneous deformation and is currently visiting in City University of Hong Kong for half year. These new and complete results in terms of the quantitative analyses for the size distribution of the grains and the precipitates will be presented in details in another paper in future.

For this paper, we have made some preliminary measurements. As shown in the Figures below, the size distribution of the grains and precipitates are shown below. Figure (a) shows the grain size distribution of CRAA alloy, collected from the EBSD images. Although diameters of 82.8% grains are lower than 2.6 μm and only 2% grains are larger than 10 μm , the volume fraction of the coarse-grained regions can reach up to 12.2%. Figure (b) shows the average diameters of coarse grains and fine grains.

Due to the significant difference between the sizes of coarse and fine L_{12} precipitates, it is difficult to characterize these coarse and fine L_{12} precipitates at the same time. So in Figure (c), the average diameter of the coarse precipitates is calculated from the STEM-EDS maps, while the average diameter of the fine precipitates is calculated from the APT results.

Since all information below is still very preliminary, please allow us not to be included in the current paper. Thanks.

The size distribution of the grains and precipitates

6. Page 8, lines 207-208, “the average spacing of the fractured sample between SFs is 48.6 nm ($4.34 \times 10^{14} \text{ m}^{-2}$ for the density of SFs) (Fig. 4a), which is evidently lower than that of the CRAA alloy before tensile testing”: can the authors elaborate the reason why the average spacing between SFs was increased after tension? Is the reduction of SF-spacing in deformed microstructure related to the strengthening mechanism or the ductilization mechanism?

Reply: Thanks a lot. According to the literature [S.W. Wu, et al. Enhancement of strength-ductility trade-off in a high-entropy alloy through a heterogeneous structure. *Acta Materialia* 165 (2019) 444-458], the strength increases with decreasing SFs spacing can be described in terms of the Hall-Petch (HP) relation:

$$\Delta\sigma = K_{\text{HP}}/\lambda^{1/2}, \quad (4)$$

where K_{HP} is the HP strengthening coefficient for SFs ($\sim 34 \text{ MPa} \cdot \mu\text{m}^{1/2}$), and λ is the SF spacing. Hence, the decrease of average SF-spacing from 74 nm to 48 nm introduce 29 MPa, which can be ignored compared with total strengthening effect. However, it provides 15 % dynamic strengthening effect, which can leading to a steady and continuous plastic deformation to large strains. So as far as we’re concerned, the underlying mechanisms responsible for the superior strength are originated from the combined effect of dislocations proliferation, HDI strengthening as well the nano-precipitation strengthening, while the decrease of average SF-spacing play a positive effect on the ductilization of the CRAA alloy.

7. The caption to Fig. 4 is missing.

Reply: Thanks a lot. The caption was missed during the format conversion of the manuscript. The caption will be checked in the new version.

If there is anything that we should do further, please do not hesitate to inform us. Thanks.

X. H. Du, Professor, Shenyang Aerospace University, Shenyang, China

Jacob C. Huang, Chair Professor, City University of Hong Kong, Hong Kong

Reviewers' Comments:

Reviewer #3:

Remarks to the Author:

I really like this paper which I believe "cuts new ground" in the HEA/MEA journey. The authors have provided a more than adequate, indeed comprehensive, response to my suggestions and have revised their manuscript accordingly. With respect to the two other reviewers, I tend to agree with you - their points in large part refer to novelty issues, but this is often the case with HEAs/MEAs issues. However, it is unrealistic to believe that this new class of multiple principal element alloys will always involve a brand new series of mechanisms of deformation, i.e., will reinvent the physics involved - of course, many of these mechanisms have been seen previously in traditional alloy systems - what else would you expect? - to my mind, the novelty in these alloys, and specifically in this paper, is that they are seeming more effective - they "provide a bigger bang for the buck" - and further that many of these mechanisms act in concert in these alloys with a synergy of mechanisms. However, whatever you believe, I feel that the authors have provided comprehensive responses to all the points raised, and in my humble opinion, this paper should be accepted for publication.

Point by point response to reviewer

Manuscript Ref. No. NCOMMS-19-539430-T

Title: Dual heterogeneous structures lead to ultrahigh strength and uniform ductility in a Co-Cr-Ni medium-entropy alloy

Journal: Nature Comm

05 April 2020

Dear Reviewer:

Thank you very much. The point to point replies are described below.

REVIEWERS' COMMENTS:

Reviewer #3 (Remarks to the Author):

I really like this paper which I believe "cuts new ground" in the HEA/MEA journey. The authors have provided a more than adequate, indeed comprehensive, response to my suggestions and have revised their manuscript accordingly. With respect to the two other reviewers, I tend to agree with you - their points in large part refer to novelty issues, but this is often the case with HEAs/MEAs issues. However, it is unrealistic to believe that this new class of multiple principal element alloys will always involve a brand new series of mechanisms of deformation, i.e., will reinvent the physics involved - of course, many of these mechanisms have been seen previously in traditional alloy systems - what else would you expect? - to my mind, the novelty in these alloys, and specifically in this paper, is that they are seeming more effective - they "provide a bigger bang for the buck" - and further that many of these mechanisms act in concert in these alloys with a synergy of mechanisms.

However, whatever you believe, I feel that the authors have provided comprehensive responses to all the points raised, and in my humble opinion, this paper should be accepted for publication.

Reply: Thank you very much.

Best regards,

J. C. Huang

Chair Professor, City University of Hong Kong